# Learning to Compare Longitudinal Images

**Heejong Kim**[1]                                                                        HEK4004@MED.CORNELL.EDU

**Mert R. Sabuncu**[1,2]                                                               MSABUNCU@CORNELL.EDU
[1] *Weill Cornell Medicine*
[2] *Cornell University*

## Abstract

Longitudinal studies, where a series of images from the same set of individuals are acquired at different time-points, represent a popular technique for studying and characterizing temporal dynamics in biomedical applications. The classical approach for longitudinal comparison involves normalizing for nuisance variations, such as image orientation or contrast differences, via pre-processing. Statistical analysis is, in turn, conducted to detect changes of interest, either at the individual or population level. This classical approach can suffer from pre-processing issues and limitations of the statistical modeling. For example, normalizing for nuisance variation might be hard in settings where there are a lot of idiosyncratic changes. In this paper, we present a simple machine learning-based approach that can alleviate these issues. In our approach, we train a deep learning model (called PaIRNet, for Pairwise Image Ranking Network) to compare pairs of longitudinal images, with or without supervision. In the self-supervised setup, for instance, the model is trained to temporally order the images, which requires learning to recognize time-irreversible changes. Our results from four datasets demonstrate that PaIRNet can be very effective in localizing and quantifying meaningful longitudinal changes while discounting nuisance variation. Our code is available at https://github.com/heejong-kim/learning-to-compare-longitudinal-images.git.

**Keywords:** Self-supervised Learning, Longitudinal Analysis, Pairwise Comparison

## 1. Introduction

Characterizing and quantifying changes over time is a fundamental goal in bio-medicine. The longitudinal imaging paradigm, where each individual is scanned at multiple time-points, is a primary approach for studying temporal dynamics.

There are two scenarios where longitudinal imaging is mainly utilized. The first is to localize and quantify change at an individual level (e.g. monitoring tumor growth and spread in a brain cancer patients (Konukoglu et al., 2007)). The second is to characterize population-level longitudinal changes so as to gain insights into biological processes (e.g. in Alzheimer's disease or aging studies (Fox et al., 1996; Fjell et al., 2009)).

In a longitudinal imaging study, whether it is at the individual or population-level, the core challenge is disentangling nuisance variation (e.g., due to differences in imaging parameters or artifacts) from meaningful (e.g. clinically significant) temporal changes (e.g., atrophy or tumor growth). Virtually all existing algorithmic approaches rely on pre-processing the data to help with this core challenge. In pre-processing, the objective is to minimize nuisance variation. A common step in pre-processing, for instance, is image registration,

where the longitudinal images are spatially normalized to some standard reference coordinate frame. The image registration step can account for orientation differences and/or other geometric changes that can be confounding - such as distortion. Even after pre-processing, there is likely residual nuisance variation that will need to be discounted and this is where statistical or machine learning-based techniques are often used. As we discuss below, these analyses can be constrained by modeling choices. For example, population-level statistical methods are not designed for characterizing longitudinal changes at the individual-level.

In this paper, we present a simple learning-based approach as a novel way for analyzing longitudinal imaging data. Crucially, our approach does not require pre-processing, is model-free, and leverages population-level data to capture time-irreversible changes that are shared across individuals, while offering the ability to visualize individual-level changes. Our core assumption is that nuisance variation, such as changes in orientation or image contrast, are independent of time, when examined across the population. We train a deep learning model, called Pairwise Image Ranking Network (PaIRNet), to compare pairs of longitudinal images. We consider two possible tasks. In the first task, which we refer to as *supervised*, PaIRNet is trained to predict a target variable that captures meaningful temporal change - such as tumor volume difference. In the second task, which we call *self-supervised*, the model is trained to predict the temporal ordering of the input longitudinal image pair. In the self-supervised setting, the only additional information we need is the timing of the images, which is often readily available. In our experiments, we demonstrate that the proposed strategy can localize and quantify individual-level longitudinal changes, even in the self-supervised setting.

## 2. Related Works

**Individual-level Longitudinal Change Detection.** Localizing and quantifying significant change is a core objective of a longitudinal study. Classical approaches, such as (Smith et al., 2001) rely on pre-processing steps such as image registration, as in (Reuter et al., 2012) and segmentation, as in (Smith, 2002). After pre-processing, a simple strategy might be to just overlay the images and study differences, such as pixel-level changes and the size of an anatomical structure. However, these differences can still contain spurious effects, due to artifacts or pre-processing issues. One can rely on statistical analyses to separate out spurious and clinically-significant changes, as demonstrated in (Nguyen et al., 2018). A weakness in these individual-level analyses is that they often do not leverage population data to better characterize and detect longitudinal change. Model-based methods mitigate this weakness by adopting mechanisms for population and individual-level dynamics (Sophocleous et al., 2022), yet these are limited by the model's capacity to capture the signal.

**Statistical Analysis of Population-level Longitudinal Change.** Another objective is to characterize temporal changes due to processes like development, aging, or disease progression, that is shared across individuals. These population-level longitudinal studies, often produce a shared map of temporal change. These studies also demand pre-processing steps to suppress nuisance variation. The classic approach, in turn, relies on statistical techniques like mixed effects models (Bernal-Rusiel et al., 2013) to reveal non-spurious temporal associations, as demonstrated in (Huang et al., 2022; Wang and Guo, 2019; Peters et al., 2019; Durrleman et al., 2013). Although widely adopted, population-level statistical analy-

ses have at least two challenges. Pre-processing and extracting measurements often require human input (annotations, quality control, etc), which can be costly and hard to achieve at scale. It can also be difficult to decide what time-varying variable to examine associations with. For example, chronological age is a common variable for studying aging-associated changes. Yet, the aging process can vary significantly across individuals, compromising the interpretation of the results.

**Machine Learning (ML) Methods for Longitudinal Images.** In recent years, ML methods have been used for longitudinal medical imaging. One approach leverages the supervised paradigm and trains a model to predict clinical trajectories from pairs of images, as in (Bhagwat et al., 2018). To derive the target label to train on, these techniques require meta-data, such as disease severity scores, which can be unreliable or unavailable. An alternative is the self-supervised paradigm, which does not require ground-truth labels. This approach has recently been applied to longitudinal imaging (Zhao et al., 2021; Ouyang et al., 2022; Couronné et al., 2021; Ren et al., 2022). The temporal ordering is used to regularize the learning of representations extracted from images, which is trained to capture inter-subject variability and within-subject temporal changes. These techniques rely on a reconstruction loss for regularization, which can make them sensitive to nuisance variation such as orientation or contrast changes over time. Thus these methods require pre-processing of the image data and cannot handle misaligned images. In contrast, the learning paradigm we propose in this paper is trained on a single task (supervised or self-supervised) and does not depend on any pre-processing. In the self-supervised setting, the ML model learns to rank an input image pair based on temporal ordering.

**Pairwise image ranking.** Pairwise image ranking is an important problem in computer vision with many applications, such as assessing aesthetic preference (Lee and Kim, 2019), skill (Doughty et al., 2018), perceptual image-error (Prashnani et al., 2018), and street safety (Dubey et al., 2016). Most of these prior works use a shared feature extraction backbone (also called a Siamese architecture). Building on the seminal work of RankNet (Burges et al., 2005) and its variant, DirectRanker (Köppel et al., 2019), we apply a similar model, for the first time, to longitudinal medical images and demonstrate how this approach can yield individual-level detection and quantification of temporal change.

## 3. Pairwise Image Ranking Network

Let us consider the problem of comparing a pair of longitudinal images. We implement a neural network called PaIRNet (for Pairwise Image Ranking Network), depicted in Figure 1. We use the convolutional backbone of ResNet-18 (He et al., 2016) for our feature network, which includes residual blocks followed by a global average pooling layer. The output is a 512-dimensional feature vector. Note that PaIRNet accepts two longitudinal images $(I_1, I_2)$ obtained from the same subject. The feature extractor network $\mathbf{f}$ is shared for the two images (which is sometimes called a Siamese architecture). The feature vectors, $\mathbf{f}(I_1)$ and $\mathbf{f}(I_2)$ are in turn subtracted and fed to a fully-connected (FC) linear layer that has no bias term. The output is then $R(I_1, I_2) = \mathbf{w}^\top(\mathbf{f}(I_1) - \mathbf{f}(I_2))$, where $\mathbf{w}$ are the weights of the FC layer and $^\top$ denotes vector transpose. For a binary classification task, this output is passed through a sigmoid unit $\sigma$ to produce a probability.

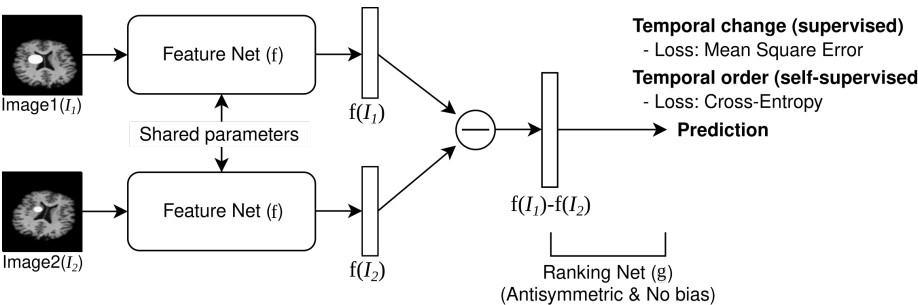

Figure 1: Pairwise Image Ranking Network (PaIRNet) consists of a feature extraction network, followed by a subtraction, and a ranking layer.

We consider both regression (supervised) and binary classification (self-supervised) tasks. In the supervised set-up, PaIRNet is trained using l2-loss to predict the change in a clinically meaningful variable $y$ such as disease severity, tumor volume difference, or chronological age. In the self-supervised set-up, PaIRNet learns to tell the temporal ordering of the input images, where we use cross-entropy loss. Further implementation details are in Appx. B.

One can show the following properties are satisfied by PaIRNet:

(a) Reflexivity: if $I_i = I_j$, then $R(I_i, I_j) = 0$ and $\sigma(R(I_i, I_j)) = 0.5$;

(b) Antisymmetry: $R(I_i, I_j) = -R(I_j, I_i)$ and $\sigma(R(I_i, I_j)) = 1 - \sigma(R(I_j, I_i))$; and

(c) Transitivity: if $R(I_i, I_j) \geq 0$ and $R(I_j, I_k) \geq 0$, then $R(I_i, I_k) \geq 0$. Similarly, if $\sigma(R(I_i, I_j)) \geq 0.5$ and $\sigma(R(I_j, I_k)) \geq 0.5$, then $\sigma(R(I_i, I_k)) \geq 0.5$.

We note that these properties are intrinsic to the tasks we consider. For the cross-sectional supervised regression setting, the task is to predict the change in the target variable $y_i - y_j$, from input image pair $(I_i, I_j)$. If $y_i \geq y_j$ and $y_j \geq y_k$, then $y_i \geq y_k$, which gives us the transitivity property.

### 3.1. Localizing Longitudinal Change

The class activation map (CAM) (Zhou et al., 2016) is a popular way to probe image classifiers trained only on image-level labels to localize objects of interest. Building on this idea, we propose a modified CAM approach to visualize the trained PaIRNet models. Given an input ordered image pair $(I_1, I_2)$, we visualize the activations at the final convolutional layer of the feature extraction network $f$ by taking a weighted sum of the activation maps. The weight of the $c$'th channel is set equal to $|\mathbf{w}_c(\mathbf{f}_c(I_1) - \mathbf{f}_c(I_2))|$.

## 4. Experiments

### 4.1. Experimental setup

#### 4.1.1. Dataset

We used two synthetic datasets and two real-world datasets to explore the utility of the proposed approach. All four datasets have sets of images that temporal change is monotonic and irreversible. We used $0.6/0.2/0.2$ ratio for the train/validation/test split, except for the Starmen. The split is done at the subject level. Figure 4 in Appx. A shows some representative examples of each dataset.

**Starmen.** The public *synthetic* Starmen dataset has been used for evaluating longitudinal frameworks (Bône et al., 2018; Couronné et al., 2021). Raising the left arm is the only temporal change within a subject and parameterized individually in the original dataset. The dynamics of change is characterized by an affine reparametrization of time as follows: $t^* = \alpha * (t - \tau)$, where $\alpha$ and $\tau$ are subject-specific parameters. For additional subject variability, we randomly rotated (uniform between -10 and 10 degrees) and translated with (-6.8,6.8) pixels. Out of 1000 sets of 10 longitudinal images, 400 sets of images are used for training, 100 sets for validation, and the rest for testing. The ground-truth progression value $t^*$ is ranged in $-0.12 \pm 4.25$.

**Tumor.** A *synthetic* tumor growth dataset was created using real T1-weighted brain MRIs from 72 healthy subjects in OASIS-2 (Marcus et al., 2010). The images were preprocessed (brain extraction and intensity normalization) using in-house software and co-registered using (Avants et al., 2009) to add synthetic longitudinal change and nuisance effect. Tumors were synthesized as discs, randomly placed within the brain tissue of the mid-axial slice. The tumor intensity, initial size, and speed of growth were randomly selected. We generated 3-5 time points per subject. In total, 292 synthetic images with an average tumor size of $14.74 \pm 6.86$ pixels were generated. All images were randomly rotated and translated with (-10, 10) degrees/pixels.

**Embryo.** We used the *real* embryo development dataset collected using time-lapse imaging incubators in an IVF study (Gomez et al., 2022). The images of developing embryos are fully annotated with 16 development phases. We used embryos with multiple phases and selected one image per phase. In total, 7784 images from 698 embryos were used. The embryos have on average $12.23 \pm 2.41$ phases.

**Aging Brain.** Structural MRI images from OASIS-3 (LaMontagne et al., 2019). In total, 754 T1-weighted images from 272 healthy subjects ranging in age from $42 - 86$ years (with a mean and std of $64.34 \pm 8.53$) at the initial scan were used. The subjects have $2.77 \pm 0.96$ time-points. We only use the mid-axial slice of linearly aligned images (Avants et al., 2009).

#### 4.1.2. Baseline methods

A straightforward baseline is to train a model to predict the target variable (e.g. tumor size or embryo phase) directly from a single image, treating all longitudinal data as i.i.d samples. We call this cross-sectional supervised regression (CSR) and present it as a baseline method, where we use the same ResNet-18 backbone and implementation details as PaIRNet (without the Siamese structure). The other baselines are the Longitudinal self-supervised learning with cosine loss (LSSL-CL; Zhao et al. (2021)) and the longitudi-

nal variational autoencoder (LVAE; Couronné et al. (2021)). Both of these methods have been successfully applied to learning latent representations from longitudinal imaging data. They rely on an image reconstruction loss with an autoencoder style architecture, in order to learn comprehensive representations that capture the full anatomical detail. Furthermore, both papers aim to disentangle temporal changes in the learned representations from time-invariant inter-individual variations. We followed the original paper implementations in our experiments and visualized time-dependent components of the learned representations. Note that, neither of these state-of-the-art baselines are well-equipped to deal with nuisance variation across different time-points and depend on pre-processing steps such as registration to suppress this variation.

## 4.2. Quantifying Longitudinal Change

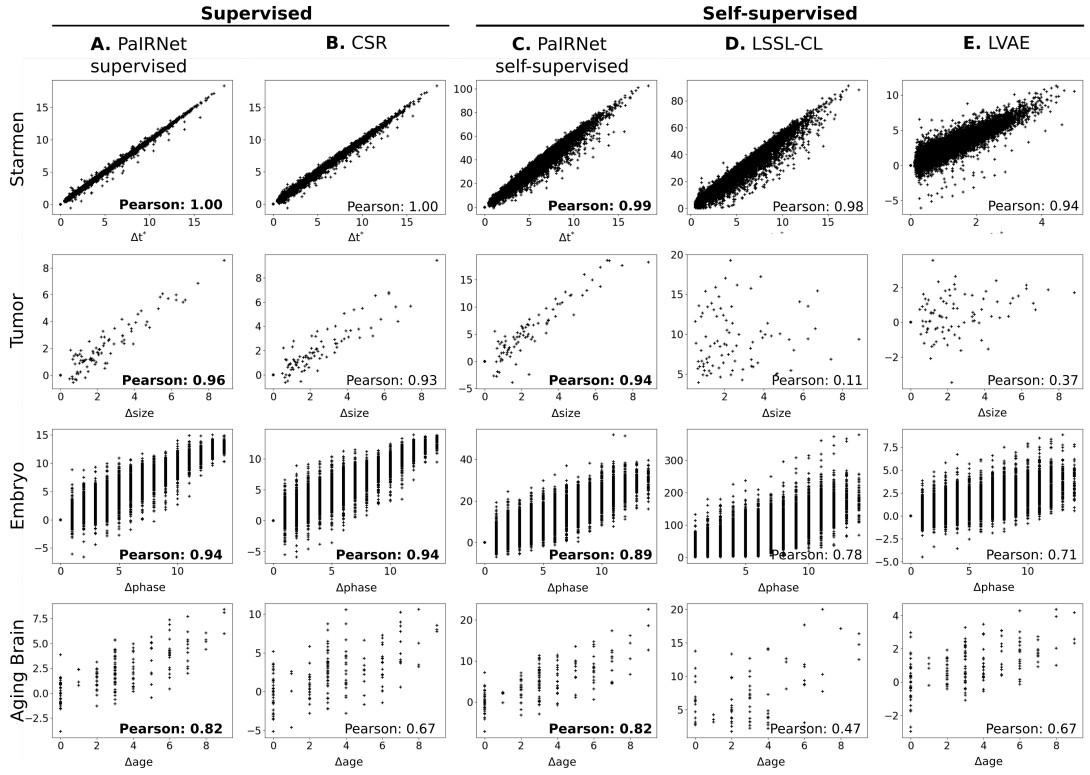

Figure 2: Correlation between ground-truth change in target variable and predicted change. Each row shows a different dataset and each column shows a different method. The coefficients in bold text are the best models for each task.

In our first analysis, we are interested in assessing how well PaIRNet and the baseline methods quantify meaningful longitudinal change. For each of the datasets, we have a target variable of interest (position of left arm in "Starmen", size of disc in "Tumor", development phase in "Embryo", and chronological age in "Aging Brain."). For each model, we can

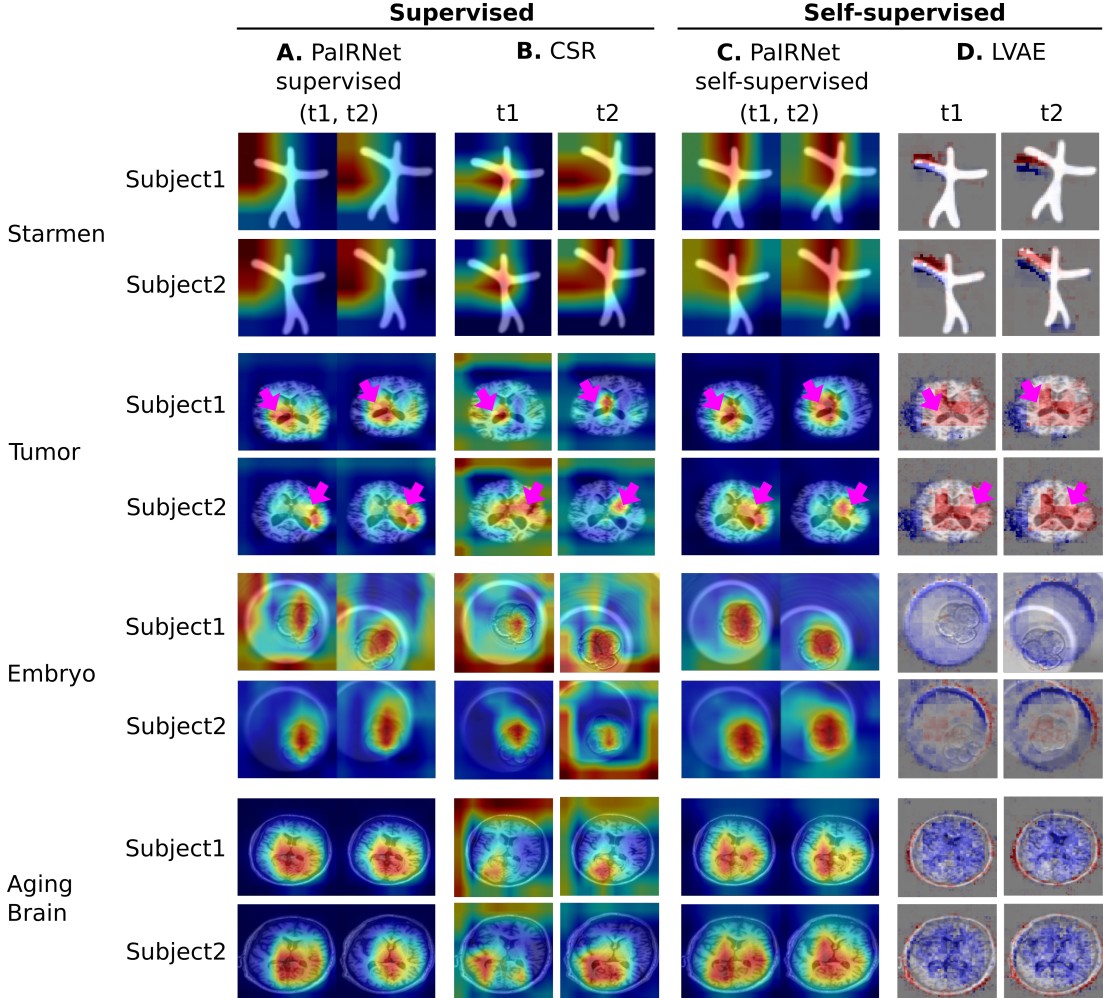

Figure 3: Localization results. Three time points from two subjects are visualized for each dataset. While cross-sectional supervised regression (CSR) and longitudinal variational autoencoder (LVAE) are affected by the subject-wise nuisance changes, PaIRNet discount those and locate the meaningful changes.

compute the predicted difference in the target variable. Note that, for the cross-sectional CSR model, this involves computing two separate predictions and then taking the difference.

Figure 2 shows scatter plots of predicted change versus ground-truth change, along with Pearson correlation coefficients (Benesty et al., 2009) that quantify the agreement. We used only ordered image pairs so that the ground-truth change is non-negative. For the self-supervised PaIRNet, we used the pre-sigmoid output values. For the LSS-CL baseline, we used the magnitude of the difference vector for the time-varying representations.

Overall, both supervised and self-supervised PaIRNet predictions showed strong correlation with ground-truth change. In particular, supervised PaIRNet (Figure 2A) yielded a

correlation that was stronger than or equivalent to CSR (Figure 2B). We suspect this is because CSR's task is harder as it needs to learn to characterize and suppress inter-subject variation. Besides, PaIRNet merely needs to learn how to compare images of the same subject. This relative difference in task difficulty is further highlighted in our supplementary analysis where we observe that supervised PaIRNet outperforms CSR with smaller training sets on the Starmen task (see Figure 5 in Appx.). Furthermore, CSR and supervised PaIRNet achieve the same correlation for the Starmen and Embryo datasets (Figure 2). In these two datasets, the longitudinal change (including the nuisance variation) is larger than inter-subject variation. For the brain aging and tumor datasets, on the other hand, there is significant inter-subject variation and supervised PaIRNet outperforms CSR.

Self-supervised PaIRNet (Figure 2C) significantly outperformed state-of-the art self-supervised approaches of LSSL-CL (Figure 2D) and LVAE (Figure 2E). Intriguingly, self-supervised PaIRNet's predicted correlations are close to that of supervised PaIRNet's (which is also the best results), suggesting that a self-supervised learning can be effective to quantify longitudinal change, even though the task itself is merely binary temporal ordering.

### 4.3. Visualizing Longitudinal Change

Next, we examine the different models' capability of localizing longitudinal change. We computed a weighted activation map as described in Section 3.1 for the PaIRNet and CSR models. Note that the maps of the pair are individually calculated for the baseline models. For the LVAE model, we followed the original paper's method, which visualizes the gradient direction (positive values in red, negative in blue) of the latent temporal feature. There is no implementation for visualizing the longitudinal changes for the LSSL-CL model.

Figure 3 illustrates the longitudinal change maps for representative examples in the four datasets from different models. The PaIRNet models, especially the self-supervised ones, seem to largely capture meaningful longitudinal changes while disregarding nuisance variation such as inter-subject variation or shift in location. Meaningful changes include the rising arm (in Starmen), synthetic tumor, and embryo cells. For Aging Brain, the maps highlight the posterior brain, including the ventricles, which is known to enlarge in aging. In contrast, the CSR models capture spurious variability, attending to areas outside the location where significant longitudinal change is happening. In the Embryo examples, CSR maps emphasize regions outside of the embryos - likely due to the change in location. While the LVAE baseline can capture the rising left arm in the Starmen dataset (which exhibits little inter-subject variation), it does a poor job on the remaining datasets. We also note that the PaIRNet maps for the Starmen dataset are not as sharply focused on the left arm as the LVAE baseline and this is something we are keen to explore in future work. Qualitatively, self-supervised PaIRNet's localization results are also as good as supervised PaIRNet's, underscoring the utility of the proposed self-supervised temporal ordering task. Further supplemental results including animation of longitudinal change maps in different magnitudes of rotation and translation are available at https://heejongkim.com/pairnet-midl.

## 5. Conclusion

We propose a simple image-pair based learning approach to capture meaningful longitudinal changes. We implement a Siamese architecture, called PaIRNet, which is trained either as

supervised (to predict change in a ground-truth variable) or self-supervised (to identify the temporal ordering). Assuming that nuisance changes are not correlated with time, our approach does not require preprocessing and learns to capture temporal changes shared across individuals while providing visualizations of individual-level changes. Experiments in four 2D image datasets show that PaIRNet can quantify and localize longitudinal change. While our results are promising, there are some drawbacks. First, the proposed approach cannot disentangle multiple temporal processes. Second, the self-supervised model cannot capture non-monotonic temporal changes. Third, there might be better architectural choices, such as a transformer that incorporates a cross-image attention mechanism. Fourth, we need to demonstrate feasibility and efficacy for 3D volumes. Lastly, we are interested in improving the visualization, e.g. based on the full-resolution CAM (Belharbi et al., 2022) that adds an additional parametric decoder (Parmar et al., 2018). We plan to explore these directions in the near future. We believe the simplicity of PaIRNet will make it practical for various longitudinal imaging analysis tasks.

## Acknowledgments

This work was supported by NIH, United States grant R01AG053949, the NSF, United States NeuroNex grant 1707312, and the NSF, United States CAREER 1748377 grant. We thank Minh Nguyen, Batuhan Karaman and Zijin Gu for useful input.

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

## Appendix A. Dataset

Figure 4 shows examples of four datasets. Note the nuisance variation is different in each dataset. The position of the left arm in Starmen, the size of the tumor in Tumor, the phase of cells in Embryo and the structural change of the brain in Aging Brain are the meaningful temporal change. Other changes are the nuisance variation.

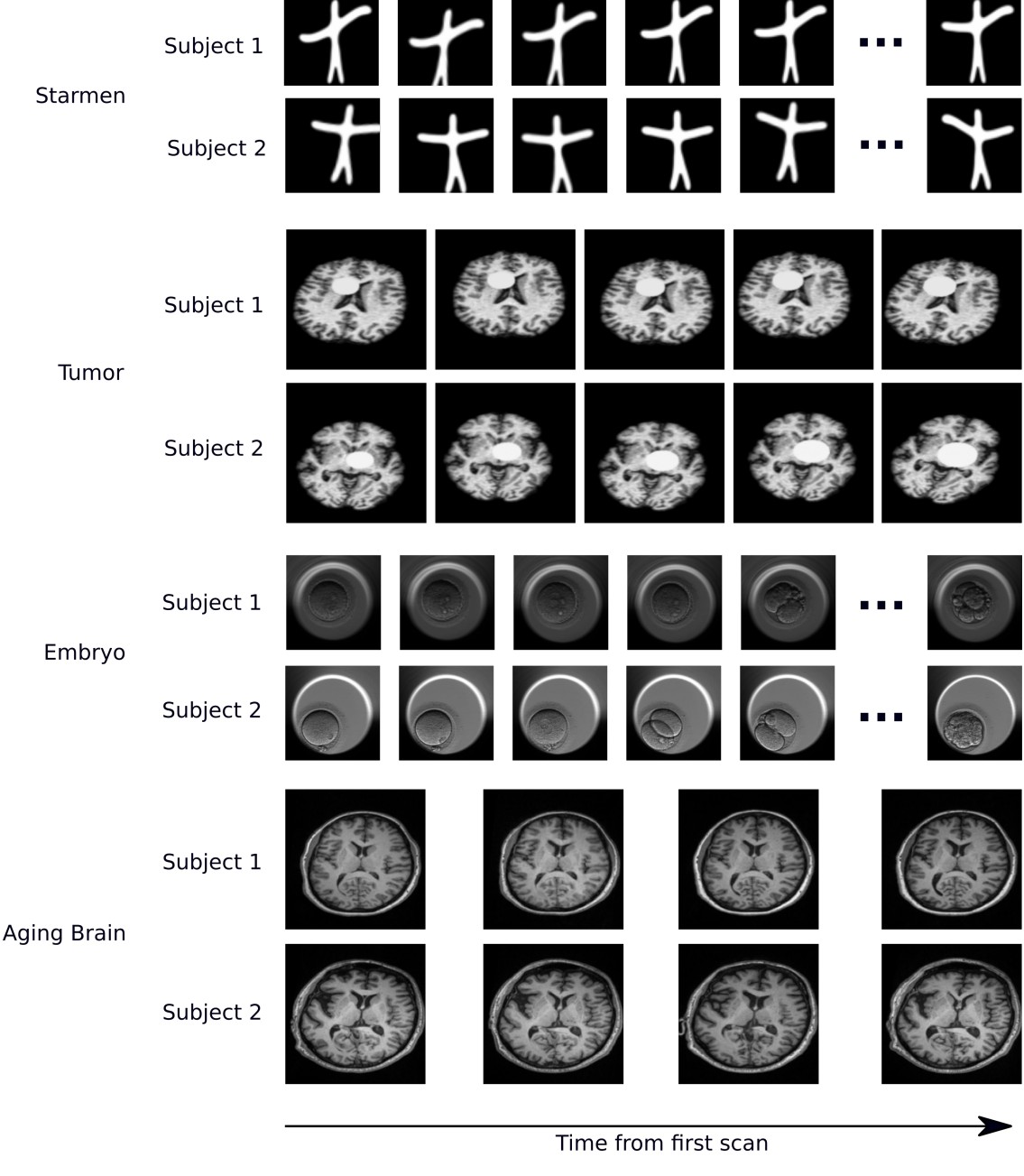

Figure 4: Representative examples of datasets.

## Appendix B. Implementation details

We used ResNet-18 (He et al., 2016) as a base network of PaIRNet and cross-sectional supervised regression (CSR). PairNet models are trained with cross-entropy loss, Adam optimizer (Kingma and Ba, 2014), and the early stopping criterion that is 5 epochs without a decrease of validation loss with a batch size of 64. For supervised models, mean squared error is used. A grid search is carried out for a learning rate using the following values: $lr = \{10^{-1}, 10^{-2}, 10^{-3}, 10^{-4}, 10^{-5}\}$. The learning rate with the best validation loss is used for testing. Table 1 shows the best learning rate.

Table 1: Learning rate.

|  | Starmen | Tumor | Embryo | Aging Brain |
|---|---|---|---|---|
| PaIRNet supervised | 0.01 | 0.01 | 0.001 | 0.01 |
| PaIRNet self-supervised | 0.001 | 0.0001 | 0.001 | 0.001 |
| cross-sectional supervised regression | 0.01 | 0.01 | 0.001 | 0.01 |

## Appendix C. Effect of Training Data Size for Quantifying Longitudinal Change.

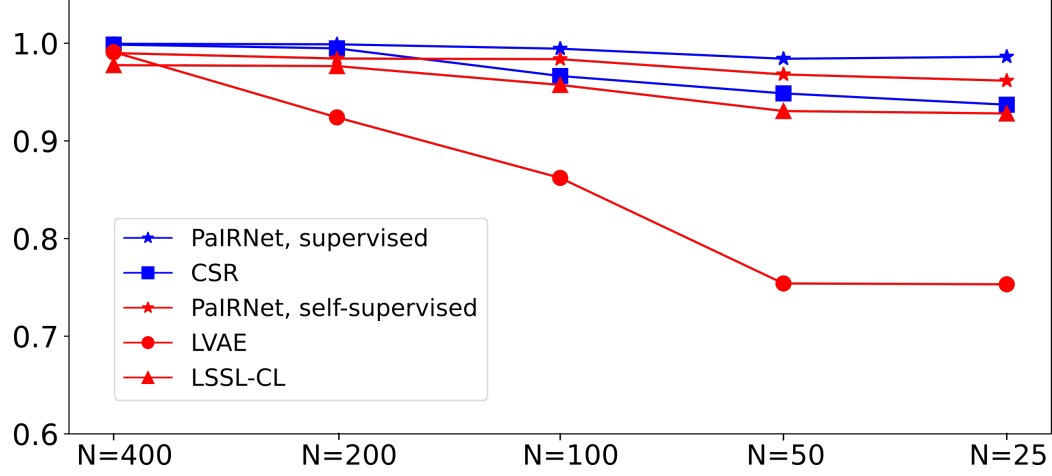

Figure 5: Correlation between ground-truth change in target variable ($t^*$) and predicted change in models trained in the different training sizes using Starmen dataset.

In this section, we show the performance of PaIRNet and baseline methods depending on the training data size for the Starmen task. We evaluated the change quantification as in Section 4.2 using Pearson correlation coefficient calculated between ground-truth change and predicted change. Figure 5 shows the coefficient results in varying data sizes. Overall,

both supervised and self-supervised PaIRNet outperform CSR and self-supervised baselines, especially with smaller data sizes.

## Appendix D. PaIRNet classification performance

In addition to the results of localization and correlation in Section 4.2 and 4.3, we compared the self-supervised PaIRNet's ranking performance with the cross-sectional supervised regression (CSR). At inference of CSR, the difference of estimated values are used to decide the ranking of a given pair. Pairs with the same ranking, which means the pairs have the same temporal label, are not used. With this, the task becomes a binary classification. The AUC result in Table 2 shows the PaIRNet we can learn the order properly with high accuracy (AUC> 0.95). Although mean square error (MSE) of regression task (CSR-MSE in Table 2 is reasonably low considering standard deviation of ground-truth labels, the ranking using the CSR results in Tumor, Embryo and Aging Brain datasets underperform compared to PaIRNet results. This demonstrates that pairwise comparison may be more suitable task for learning difference between images than than CSR. The results from Aging Brain dataset highlights this. From the high CSR MSE, we can determine predicting age from images is more difficult tasks than others. However, PaIRNet AUC was 0.985, which is comparable to other datasets.

Table 2: AUC result of binary classification from testing sets. Pairwise Imaging Ranking Network (PaIRNet) outperforms cross-sectional supervised regression (CSR) results. Although mean square error (MSE) loss of CSR is low, the AUC was not as high as PaIRNet.

|                         | Starmen | Tumor | Embryo | Aging Brain |
|-------------------------|---------|-------|--------|-------------|
| PaIRNet self-supervised | **1.0** | **0.948** | **0.994** | **0.985** |
| CSR                     | **1.0** | 0.929 | 0.965  | 0.893       |
| CSR-MSE                 | 0.017   | 1.239 | 1.355  | 40.215      |

## Appendix E. Localization performance of Tumor task

In this section, we show the quantitative result of localization of using Tumor dataset. Dice similarity coefficient is utilized as the metric. We binarized the activation masks using different thresholds in the range between 0.6 and 0.9. Figure 6 demonstrates the average DSC between the ground-truth and binarized activation masks using varying thresholds. Overall, the PaIRNet models outperform the CSR model. As the activation maps gives the most discriminative part, not the whole object, the DSC score is low for all models. However, the DSC score can be used to confirm if the model locate the correct region. Figure 3B Subject 2's t1 example shows the case that the activation map highlights the distant regions. For that map, the binarized result will have the predicted tumor at the left corner, not in the brain. Thus, this DSC result can quantitatively support that PaIRNet models localize longitudinal change better than the CSR model.

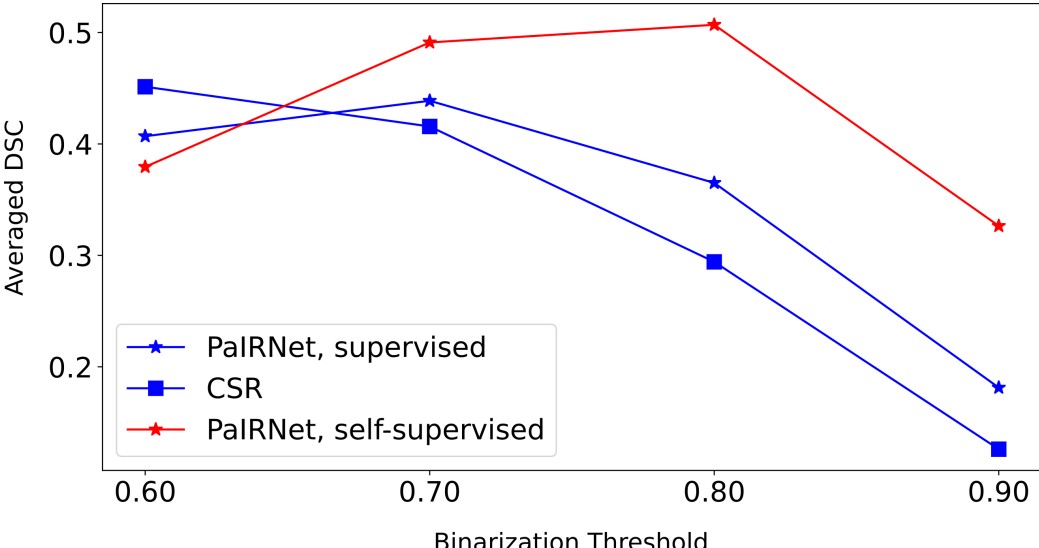

Figure 6: Averaged Dice Similarity Coefficient (DSC) between the target mask and the binarized activation map.

## Appendix F. Additional Evaluation on Modified Datasets

We ran additional evaluation on the modified datasets to further address that PaIRNet does not require preprocessing. As Embryo images already has all image variances (translation, rotation, and intensity), we did not consider Embryo for this evaluation.

### F.1. Dataset

**Starmen.** Using the same Starmen dataset with images augmented using translation and translation (described in Section 4.1.1), we randomly changed brightness and contrast with a factor of (0.8, 1.2).

**Tumor.** We re-generated the synthetic tumor dataset using the raw images, instead of the preprocessed scans. After placing the synthetic tumors, all images were randomly rotated, translated with (-10, 10) degrees/pixels, and a random perturbation was applied to the brightness and contrast of images with a factor of (0.8, 1.2).

**Aging Brain.** We used un-preprocessed images (i.e., no skull stripping or intensity normalization was applied) described in Section 4.1.1. We added random rotation (-10, 10 degrees), translation (8.8, 12.8 pixels), brightness (0.8, 1.2 factor) and contrast (0.8, 1.2 factor). to simulate nuisance variation.

### F.2. Quantifying Longitudinal Change

Using the updated datasets, we trained the PaIRNet and obtained the Pearson Correlation Coefficient results in Section 4.2. We followed the same implementation details in Section B. Table 3 illustrates the agreement between the ground-truth change and the pre-

dicted change. While some of the numerical values were altered after these changes, the overall pattern of results and conclusions remain the same. The proposed PaIRNet strategy is robust to such nuisance variation and learns to ignore it, as long as it is independent over time. The independence assumption is satisfied for the random augmentations implemented during training. This applies to both the supervised and self-supervised versions of PaIRNet.

Table 3: Correlation between ground-truth change in target variable and predicted change using modified dataset in this Section.

|                        | Starmen | Tumor | Aging Brain |
|------------------------|---------|-------|-------------|
| PaIRNet supervised     | 1.00    | 0.98  | 0.80        |
| PaIRNet self-supervised | 0.98   | 0.96  | 0.78        |

Table 4: Learning rate for the modified datasets.

|                        | Starmen | Tumor | Aging Brain |
|------------------------|---------|-------|-------------|
| PaIRNet supervised     | 0.01    | 0.001 | 0.01        |
| PaIRNet self-supervised | 0.01   | 0.01  | 0.01        |

To further demonstrate how the proposed methods can handle different types of nuisance variation, we used the Tumor dataset. In the Table 5, we show Pearson Correlation Coefficient values between ground truth and predicted change in target variable for different versions of the data perturbation, deterministically applied to the second time point of the test images. We observe that PaIRNet yields strong correlations under all scenarios, highlighting how the proposed approach can handle nuisance variation. We emphasize that, for these results, the models were trained with random augmentations (as described above), which were independent from time.

## Appendix G. Network Ablation

We computed Pearson Correlation values between predicted and ground-truth change in the target variable for different versions. In the first version, we simply considered an untrained (randomly initialized with (He et al., 2015)) model. In the second version, we froze the weights of the randomly initialized feature network and only updated the final fully connected layer during training. Table 6 shows the results of supervised and self-supervised PaIRNet (only FC trained and frozen random feature network). The results of frozen networks are far worse than the trained PaIRNet results in Figure 2.

Table 5: Correlation between ground-truth change in target variable and predicted change using additional synthetic tumor dataset. Each row shows different image transforms and magnitudes.

|  | PaIRNet Supervised | PaIRNet Self-supervised |
|---|---|---|
| Translate X: $-10$ pixels | 0.96 | 0.96 |
| Translate X: $+10$ pixels | 0.97 | 0.96 |
| Translate Y: $-10$ pixels | 0.97 | 0.97 |
| Translate Y: $+10$ pixels | 0.97 | 0.96 |
| Rotate: $-10°$ | 0.97 | 0.95 |
| Rotate: $+10°$ | 0.96 | 0.95 |
| Contrast 0.8 | 0.97 | 0.97 |
| Contrast 1.2 | 0.97 | 0.97 |
| Brightness 0.8 | 0.97 | 0.97 |
| Brightness 1.2 | 0.97 | 0.97 |

Table 6: Network Ablation. Pearson correlation coefficients between ground-truth change in the target variable and predicted change is calculated.

|  | Starmen | Tumor | Embryo | Aging Brain |
|---|---|---|---|---|
| Frozen feature network (Supervised) | 0.57 | 0.32 | 0.64 | 0.48 |
| Frozen feature network (Self-supervised) | 0.59 | 0.14 | 0.67 | 0.51 |
| Untrained PaIRNet | -0.02 | -0.12 | 0.01 | -0.15 |

