# OpenReview forum: "Learning to Compare Longitudinal Images"
_MIDL.io/2023/Conference — MIDL 2023 Oral_

### Official Review · Reviewer_qoAe · 2023-02-02

**Confidence:** 4
**Preliminary Rating:** 3

**Summary:**

The authors propose the use of a ranking network to study temporal changes in images. A ranking network consists on an embedding component f, that takes input images (x) and produces and feature vector h = f(x). Image pair rankings for (x1, x2) are obtained by passing the difference between two image features f(x_1) - f(x_2) to the ranking component of the network g. The authors propose two methods for defining the targets for training the ranking network, the temporal change (supervised) or the temporal order (self-supervised).

**Strengths:**

- The proposed method is well described and easy to understand
- Visualisations on where the proposed method is attending to in image is provided via the use of class activation maps
- 5 datasets are used to benchmark the method

**Weaknesses:**

As an application of the ranking networks proposed in (Koppel et al., 2019) and (Burges et al., 2005) to medical image data, using 3D volumes or more realistic datasets would have added more validity to the study.

**Deanonymize Review:**

no

**Detailed Comments:**

- Reference to CSR (Figure 2C), should this be 2B?


**Paper Type:**

validation/application paper

**Questions To Address In The Rebuttal:**

- What is the extension made to the work proposed in (Koppel et al., 2019) and (Burges et al., 2005)?
- The authors argue that the proposed method is better at handling "nuisance" variation, why, intrinsically is pairnet better at handling this than the baseline methods?

---

### Official Review · Reviewer_tb2h · 2023-02-04

**Confidence:** 4
**Preliminary Rating:** 4

**Summary:**

The authors present a simple Siamese network-based approach called PaIRNet to compare pairs of longitudinal images at the feature level, with or without supervision.
In the self-supervised setting, the Siamese network is trained to temporally order the images, which requires learning to recognize time-irreversible changes.

**Strengths:**

This is a well-written paper regarding clarity, formulation, and analysis.
The results from four datasets show that the proposed PaIRNet can be very effective in localizing and quantifying meaningful longitudinal changes while discounting nuisance variations.
The authors conduct extensive experiments to analyze the proposed approach.

**Weaknesses:**

I have a few concerns about the methodology (esp. the design choice) and results.

**Does the network learn invariance to nuisance variations?**
The design of the Siamese network seems not to guarantee that it would be immune to nuisance variation. Do the authors expect the network to learn such property? If yes, other regularizations, such as contrastive loss to gain rotation invariance, could be applied.

**An ablation study would better understand the results.** The results of self-supervised are interesting. I am curious about the results of random networks, i.e., if the authors use an un-trained network (randomized [1]) to extract features and then perform the co-relation analysis.

**2D or 3D?** It is not clear if the proposed approach is 2D or 3D. The authors claim that their method does not require any pre-processing. This will not hold if the method is in 2D due to the random orientation of 3D images.

**Some missing related work**. Some related works on self-supervision could be discussed, such as [2].


References

[1] Delving Deep into Rectifiers: Surpassing Human-Level Performance on ImageNet Classification, ICCV 2015
[2] Local Spatiotemporal Representation Learning for Longitudinally-consistent Neuroimage Analysis, NeurIPS 2022


**Deanonymize Review:**

no

**Detailed Comments:**

Same as above.

I have a few concerns about the methodology (esp. the design choice) and results.

**Does the network learn invariance to nuisance variations?**
The design of the Siamese network seems not to guarantee that it would be immune to nuisance variation. Do the authors expect the network to learn such property? If yes, other regularizations, such as contrastive loss to gain rotation invariance, could be applied.

**An ablation study would better understand the results.** The results of self-supervised are interesting. I am curious about the results of random networks, i.e., if the authors use an un-trained network (randomized [1]) to extract features and then perform the co-relation analysis.

**2D or 3D?** It is not clear if the proposed approach is 2D or 3D. The authors claim that their method does not require any pre-processing. This will not hold if the method is in 2D due to the random orientation of 3D images.

**Some missing related work**. Some related works on self-supervision could be discussed, such as [2].


References

[1] Delving Deep into Rectifiers: Surpassing Human-Level Performance on ImageNet Classification, ICCV 2015
[2] Local Spatiotemporal Representation Learning for Longitudinally-consistent Neuroimage Analysis, NeurIPS 2022

**Paper Type:**

both

**Questions To Address In The Rebuttal:**

In summary of the above points:

**Does the network learn invariance to nuisance variations?**

**An ablation study on random networks would help the read better understand the results of the proposed method.**

**It is method in 2D or 3D?**

---

### Official Review · Reviewer_b43M · 2023-02-04

**Confidence:** 4
**Preliminary Rating:** 4
**Recommendation:** Poster

**Summary:**

This paper addresses the problem of deep learning for longitudinal studies, and especially focussing on variances between the observations.  The authors propose a Siamese network with shared parameters to compare two images of the same study. This network can be trained supervised or self-supervised by either directly optimizing the network on the target metric or by ranking the images according to their relative timestamp, respectively. The network is evaluated on four datasets and compared with three other baseline methods in terms of correlation with the relevant metric of the longitudinal study (tumor size, aging). Besides just using quantitative metrics, the authors also use a technique to visualize the temporal changes and demonstrate, that their method better captures these changes.

**Strengths:**

- Simple and reasonable approach to the problem by using a Siamese network, resulting in better results compared to the baseline methods, especially for the self-supervised method.
- Changing distributions over time is a general and relevant problem for deep learning models. Addressing this issue in longitudinal studies by adding random rotation and translations simulates issues arising when encountering real data. To also evaluate different study subjects, the authors perform the experiments on four datasets, with better performance on each of them against the baselines.
- The Visualization method qualitatively shows the network's capability of detecting where the clinical relevant change can be found in the image. In general, the proposed network seems to better be able to localize the temporal changes than the other baseline method. This is especially relevant in the medical domain, as explainability and trustworthiness are critical targets.
- The authors are willing to publish the code to support better reproducibility. They also provide details about the training parameters.


**Weaknesses:**

- One justified critic point is, that other method rely on pre-processing and normalization. However, in the experiment sections, some datasets are also normalized (Tumor) and aligned (Aging). The dataset examples show that in terms of color variations (e.g. due to different acquisition protocols), the intra-subject variability is very low and only the inter-subject variability is high only in the case of the Embryo dataset. Therefore, these datasets can only partially support the author's claim of being able to handle  “nuisance variation” without preprocessing.
- The augmentations and perturbations of the images are limited to rotation and translation. However, especially in the medical domain, there are several other types of variances like different acquisition protocol.


**Deanonymize Review:**

no

**Detailed Comments:**

- Please higher the resolution of the images or make them as vector images
- Modify network architecture image to show supervised and unsupervised version. Maybe add the loss formulas in the image
- Use a Transpose symbol instead of an ‘
- I would recommend rewriting the conclusion and highlight the method's capabilities and advantages, and then mention the points of further improvement.


**Paper Type:**

both

**Questions To Address In The Rebuttal:**

While showing that the method is capable of handling small rotations and translation to simulate unregistered data, can it handle differences in acquisition protocol or handle un-normalized data? It is a very relevant issue and can be simulated using simple augmentations that change the image colors (you mentioned contrast multiple times).  Since one of the central arguments for the method is that it does not rely on these kinds of preprocessing steps, an evaluation is required. Considering the size of the network, this should be possible during rebuttal.


The authors were able to answer the open question by presenting the evaluation results for non-normalized and perturbed datasets, simulating stronger differences in the acquisition process. They were able to show that their proposed method is robust to these differences and still performs similarly compared to the normalized version of the dataset. This justifies the claims made by the authors that their method can handle different types of nuisance variations. They also performed an augmentation-specific ablation study, showing nearly the same performance for all augmentations. Given these new evaluations, but still the limitations of the approach, I will change the rating accordingly and wish the authors all the best on their further way.

---

### Meta-Review · Area_Chair_UyeL · 2023-02-24

**Recommendation:** Accept (Poster)
**Confidence:** 5

**Metareview:**

This paper proposes PaIRNet (Pairwise Image Ranking Network), a Siamese network with shared parameters to compare images to rank longitudinal image sequences in both supervised and self-supervised settings. The proposed method has novelty and practical value in longitudinal studies. The paper is written clearly and provides intensive validation. Critiques from reviewers are well addressed in the rebuttal, and the authors provided a revised version according to the reviewers' suggestions, including additional experiments such as ablation studies.